# Does surgical approach affect Hirschsprung-associated enterocolitis risk? A comparison between transanal Swenson-like and endorectal pull-throughs

Azzahra Fatinnuha Azmi Prayogi Putri[1], Dwiki Afandy[1], Ahmad Zakiy Habibiy[1], Setiani Silvy Nurhidayah[1], Khanza Adzkia Vujira[1], Pramana Adhityo[1], Gilang Vigorous Akbar Eka Candy[1], Kristy Iskandar[2], Eko Purnomo[3], Gunadi[1]*

1 Pediatric Surgery Division, Department of Surgery, Faculty of Medicine, Public Health and Nursing, Universitas Gadjah Mada/Dr. Sardjito Hospital, Yogyakarta, Indonesia, 2 Department of Child Health, Faculty of Medicine, Public Health, and Nursing, Universitas Gadjah Mada/UGM Academic Hospital, Yogyakarta, Indonesia, 3 Pediatric Surgery Division, Department of Surgery, Faculty of Medicine, Public Health and Nursing, Universitas Gadjah Mada/UGM Academic Hospital, Yogyakarta, Indonesia

* drgunadi@ugm.ac.id

## Abstract

Hirschsprung-associated enterocolitis (HAEC) is a complication of Hirschsprung disease (HSCR) that may occur both before or after surgery. Transanal endorectal pull-through (TEPT) is one of the standard treatments for HSCR. In addition, transanal Swenson-like pull-through (TSLPT) is a recent technique combining the Swenson method with the posterior sagittal approach. We aimed to compare the incidence of HAEC following TSLPT versus TEPT in HSCR patients and examine their association with predictor factors. We retrospectively reviewed the medical records of HSCR patients who underwent either TSLPT or TEPT at our institution between 2018 and 2023. The diagnosis of HAEC was established using an HAEC scoring system with cut-off values of 4 and 10. This study included 29 patients who underwent TSLPT and 41 who underwent TEPT. Using a cut-off of ≥4, the proportion of HAEC in TEPT and TSLPT was 24.4% and 6.9%, respectively. When applying the cut-off of ≥10, the proportion decreased to 9.8% in TEPT and 3.4% in TSLPT. There was no statistically significant difference in the incidence of HAEC between the TSLPT and TEPT groups (p = 0.056). Postoperative albumin level was statistically associated with HAEC in the TEPT group (p = 0.03), but no other predictor factors, including sex, age at definitive surgery, type of aganglionosis, nutritional status, and postoperative hemoglobin level (p > 0.05). Subsequently, multivariate analysis indicated that albumin level was significantly associated with the occurrence of HAEC in HSCR patients following TEPT (p = 0.047). In conclusion, our study suggests that the incidence of HAEC tends to be higher following TEPT compared to TSLPT. Monitoring albumin levels postoperatively may be considered as a potential risk indicator for diagnosing HAEC in patients

**Data availability statement:** All relevant data are within the manuscript and its Supporting information files.

**Funding:** The author(s) received no specific funding for this work.

**Competing interests:** The authors have declared that no competing interests exist.

undergoing surgery especially TEPT, as hypoalbuminemia showed significance only within this surgical subgroup.

## Introduction

Hirschsprung disease (HSCR) is a leading cause of functional obstruction in children, resulting from defective migration of neural crest cells during fetal development [1]. This developmental anomaly leads to the absence of ganglion cells in the affected bowel segment, impairing peristalsis and causing functional obstruction [1]. HSCR affects approximately 1 in 5,000 live births, although incidence rates vary across regions and ethnic groups [2]. In Indonesia, a recent study reported a higher incidence, estimated at 1 in 3,250 live births, highlighting the pressing need for early recognition and intervention in this population [3]. Timely diagnosis—established through a combination of clinical assessment, imaging studies, and histopathological confirmation from rectal biopsy—is crucial to prevent life-threatening complications and to guide definitive surgical management [1,4].

Hirschsprung-associated enterocolitis (HAEC) has long been recognized as a major source of postoperative morbidity and mortality in patients with Hirschsprung disease, as demonstrated in earlier studies that reported significant risk despite surgical advancement [5]. Despite progress in surgical management, HAEC continues to pose a significant risk for morbidity and mortality [5]. The incidence of HAEC ranges from 1% to 10%, with the highest risk observed in neonates before definitive surgery [5]. Patients with HSCR who develop HAEC during the neonatal period typically experience a hospital stay that is twice as long as those without this complication [6]. More recent studies, however, continue to demonstrate that HAEC is associated with longer hospitalization and increased morbidity in the current era of improved perioperative care and surgical techniques [7,8]. For example, one 2022 review noted that the mean duration of hospitalization for HAEC patients was 13 days (ranging from 6 to 29 days) and that medical management of HSCR children with HAEC is 2.5 times costlier than for those without HAEC [7]. A comprehensive review published in 2024 reinforces that HAEC is an important cause of morbidity and the leading cause of mortality in patients with HSCR [8]. These contemporary data reinforce that HAEC remains a clinically impactful complication despite advancements in management [8]. Several factors—such as timing of definitive surgery, nutritional status, and maternal characteristics—have been considered as contributors to the development of HAEC [9].

Resection of the aganglionic segment remains the cornerstone of definitive surgical intervention, with preservation of anal sphincter function as a critical objective [10]. Established procedures—such as the Swenson, Duhamel, and Soave techniques—are routinely performed; however, the need for less invasive approaches with improved postoperative outcomes has driven significant surgical innovation [3]. The transanal endorectal pull-through (TEPT) technique has been widely adopted for its minimally invasive nature, shorter hospital stays, and lower complication rates than transabdominal approaches [11,12]. In recent years, emerging techniques such as the transanal Swenson-like pull-through (TSLPT) have shown promise in reducing

mechanical and functional complications, including postoperative HAEC [13]. By allowing rectal dissection via the anal canal and limiting the retention of the aganglionic cuff to a 0.5–1 cm segment proximal to the dentate line, this method addresses potential sources of residual obstruction [14]. Recent studies suggested that the TSLPT technique may lower the risk of mechanical and functional complications, highlighting the urgency of evaluating and adopting the most effective surgical approach to improve patient outcomes, including HAEC [14,15]. Despite these developments, evidence remains limited, and the incidence of HAEC continues to vary widely across centers, underscoring the need for further investigation into factors contributing to its occurrence.

HAEC is hypothesized to arise from a multifactorial interplay of decreased enteric nervous system function, altered gut motility, dysbiosis, and dysregulation of mucosal immunity [8,16,17]. Variations in surgical techniques may influence these parameters differently. For example, differences in the extent of aganglionic segment excision, anastomotic tension, and postoperative colonic motility restoration may contribute to the varying susceptibility to HAEC between procedures [16,17]. These physiological distinctions may partly explain the differing proportions of HAEC observed between TSLPT and TEPT in our cohort.

However, there has been no direct comparison of HAEC incidence between TSLPT and TEPT techniques. Understanding this association is essential for selecting the most effective surgical approach and minimizing adverse outcomes, including HAEC. This study aimed to compare the incidence of HAEC following TSLPT and TEPT procedures and to evaluate its association with predictor factors such as sex, aganglionosis type, nutritional status, age at definitive surgery, and postoperative hemoglobin and albumin levels.

## Materials and methods

### Study design and setting

A cross-sectional study was conducted using secondary data obtained from medical records. Data on HAEC in patients who underwent TSLPT or TEPT between 2018 and 2023 were collected from medical records at our institution. The primary outcome was the occurrence of HAEC after surgery, assessed retrospectively using a scoring system based on clinical symptoms, radiological findings, and laboratory results documented in medical records.

Diagnosis of HAEC was determined using a clinical scoring system developed initially in prior literature (e.g., the Pastor score and the Frykman score), which incorporates history, physical examination, radiologic, and laboratory criteria [18,19]. In the Pastor score the original diagnostic cut-off was ≥ 10 points [18]; subsequently, other authors proposed that a lower cut-off of ≥4 points increased sensitivity [19]. In our study, we applied both cut-off values (≥10 and ≥4) to the scoring of each patient encounter to compare how the proportion of HAEC would vary under each threshold. Scoring was performed by a pediatric surgeon who extracted the presence/absence of each criterion from the medical record [20]. The criteria included: explosive diarrhea, foul-smelling diarrhea, bloody diarrhea, abdominal distention, fever, lethargy, dilated bowel loops on imaging, decreased peripheral perfusion, leukocytosis, and shift to left. We acknowledge that the scoring system has limited prospective validation and that inter-rater reliability was not formally measured in our study—this is noted in the Limitations section.

The protocol of this retrospective study was reviewed and approved by the Medical and Health Research Ethics Committee (MHREC), Faculty of Medicine, Public Health, and Nursing, Universitas Gadjah Mada, Yogyakarta, Indonesia (Approval number: KE/FK/0937/EC/2023). Written informed consent was obtained from the parents or legal guardians of all participating patients prior to inclusion in the study, permitting the use of anonymized medical data for research purposes. All data were fully anonymized before access and analysis. The study was conducted in accordance with the principles of the Declaration of Helsinki and institutional ethical guidelines.

### Eligibility criteria

The inclusion criteria comprised patients diagnosed with HSCR who had undergone TSLPT or TEPT at our institution, with complete medical records available from 2018 to 2023 [21]. Only patients with complete and accessible medical records were considered eligible for analysis during the specified period. The initial clinical suspicion of HSCR was based on

characteristic signs and symptoms, including delayed passage of the first meconium beyond 24 hours of life, progressive abdominal distension, feeding intolerance, and/or failure to thrive [22]. In older children, chronic constipation of varying severity was also documented as a frequent presentation [22]. The definitive diagnosis was confirmed by rectal biopsy, which demonstrated the absence of ganglion cells in both Meissner's and Auerbach's plexuses [23]. Patients whose medical records lacked essential diagnostic, clinical, or surgical data were excluded from the study.

## Data collection

Data were collected for research purposes between October 2, 2023 and February 2, 2024. Data were retrospectively collected from patient medical records, focusing on predictor variables relevant to postoperative outcomes, including sex, age at definitive surgery, type of aganglionosis, nutritional status, and postoperative laboratory findings, including hemoglobin and albumin levels. Authors had access to information that could identify anonymized individual participants during or after data collection. Identifying variables (such as names, dates of birth, or medical record numbers) were replaced with study codes, and only de-identified datasets were used in the statistical analysis. No identifiable information was stored, analyzed, or reported in the manuscripts. Only cases with complete medical records were included, and no imputation was performed. Because no partially missing variables were present in the included dataset, sensitivity analyses were not required.

## Study variables

HAEC were quantified using a standardized scoring system (Table 1) [18]. In the present study, cut-off values of 10 and 4 were utilized for comparison [18,19,24]. The TSLPT procedure was performed as a modification of the Swenson technique, involving complete removal of the aganglionic segment through a transanal approach, while the TEPT technique used an endorectal dissection via a transanal route. Sex was categorized as male or female. Age at definitive surgery was calculated from birth to the time of definitive surgery. Patients aged 0–1 month were classified as neonates, while those older than 1 month were considered post-neonates. Nutritional status was assessed based on body mass index (BMI) and categorized as underweight (<5th percentile), ideal (5th–85th percentile), overweight (85th–95th percentile), or obese (>95th percentile) using the WHO Chart. The type of HSCR was determined based on the length of the aganglionic segment: short- and long-segment aganglionosis [25]. A short-segment disease was defined as aganglionosis extending up to the junction between the sigmoid and descending colon. A long-segment disease involved aganglionosis located proximal to that junction, with some portion of the colon still containing ganglion cells [25]. Postoperative hemoglobin and albumin levels were based on laboratory measurements taken after surgery. Age-specific reference values for hemoglobin were as follows: newborns, 14–24 g/dL; 0–2 weeks, 12–20 g/dL; 2–6 months, 10–17 g/dL; 6 months–6 years, 9.5–14 g/dL; and 6–18 years, 10–15.5 g/dL. Hypoalbuminemia was defined as a serum albumin concentration below 3.5 g/dL.

## Sample size and sampling method

The sample size for this study was calculated using a single-proportion formula to estimate the minimum required number of subjects: $n = Z^2 \alpha PQ/d^2$, where n denotes the minimum sample size, P represents the estimated proportion of Hirschsprung disease incidence (0.2310), Q is $1 - P$, α is the significance level (0.05), Z is the corresponding Z-score (1.96), and d is the absolute precision level (10%). The proportion used for the calculation was 0.125. Substituting these values into the formula yielded: $n = (1.96)^2 \times 0.231 \times (1 - 0.231)/ (0.12)^2$, resulting in an estimated minimum sample size of 47.4, which was rounded up to 48. However, because all eligible HSCR cases with complete medical records from 2018 to 2023 were included, the final sample size was 70.

## Statistical analysis

Differences in the incidence of HAEC between patients who underwent the TEPT technique and those treated with the TSLPT technique were analyzed and presented as proportional differences. Chi-square tests were applied to compare the

**Table 1. Scoring system for Hirschsprung-associated enterocolitis based on clinical history, physical examination, radiologic findings, and laboratory parameters.**

| Characteristic | Score |
|---|---|
| **History** | |
| • Diarrhea with an explosive stool | 2 |
| • Diarrhea with foul-smelling stool | 2 |
| • Diarrhea with bloody stool | 1 |
| • History of enterocolitis | 1 |
| **Physical examination** | |
| • Explosive discharge of gas and stool on rectal examination | 2 |
| • Abdominal distension | 2 |
| • Decreased peripheral perfusion | 1 |
| • Lethargy | 1 |
| • Fever | 1 |
| **Radiological examination** | |
| • Multiple air-fluid levels | 1 |
| • Dilated bowel loops | 1 |
| • Sawtooth appearance with irregular mucosal lining | 1 |
| • Cut-off sign in rectosigmoid with absence of distal air | 1 |
| • Pneumatosis intestinalis | 1 |
| **Laboratory finding** | |
| • Leukocytosis | 1 |
| • Shift to left | 1 |
| **Total** | 20 |

proportion of HAEC between the TSLPT and TEPT groups. Chi-square tests were applied to compare categorical variables, as all contingency table cells met the minimum expected frequency required for Chi-square analysis. Multivariate analysis was conducted using logistic regression to identify independent predictors of HAEC, adjusting for potential confounders. Post-hoc analyses for categorical variables were performed using adjusted residuals following significant Chi-square tests; values ≥ |2| were considered indicative of meaningful cell-level deviation. All statistical analyses were performed using SPSS version 27.0 (IBM, Armonk, NY, USA). A *p*-value<0.05 was considered to indicate statistical significance.

## Results

### Baseline characteristics of HSCR patients treated with TSLPT and TEPT

This study included a total of 70 patients, with 29 undergoing TSLPT and 41 undergoing TEPT. The cohort was composed of 48 males (68.6%) and 22 females (31.4%). Most patients (91.4%) had short-segment aganglionosis, and 85.7% were classified as underweight—notably, 97.1% of patients presented with main complaints beyond the neonatal period. Post-operative anemia was observed in 24% of patients, while hypoalbuminemia was found in 37%. There was no significant association between these predictor factors and the type of surgical procedure performed (Table 2).

### Proportions of Hirschsprung-associated enterocolitis following TSLPT and TEPT surgery

The occurrence of HAEC following TSLPT and TEPT procedures was evaluated using two diagnostic score cut-off values: ≥ 10 and ≥4. Using a cut-off of ≥4, the proportion of HAEC in TEPT and TSLPT was 24.4% and 6.9%, respectively. When applying the stricter cut-off of ≥10, the proportion decreased to 9.8% in TEPT and 3.4% in TSLPT. There was no statistically significant difference in the incidence of HAEC between the TSLPT and TEPT groups (p = 0.056) (Table 2).

**Table 2. Baseline characteristics of Hirschsprung disease patients who underwent pull-throughs in our institution.**

| Characteristic | TSLPT (n = 29), n (%) | TEPT (n = 41), n (%) | Total (n = 70), n (%) | p-value[a] |
|---|---|---|---|---|
| **Sex** | | | | |
| • Male | 19 (65.5) | 29 (70.7) | 48 (68.6) | 0.643 |
| • Female | 10 (34.5) | 12 (29.3) | 22 (31.4) | |
| **Age at definitive surgery** | | | | |
| • Neonates | 2 (6.9) | 0 (0) | 2 (2.9) | 0.088 |
| • Post-neonates | 27 (93.1) | 41 (100) | 68 (97.1) | |
| **Nutritional status** | | | | |
| • Underweight | 24 (82.8) | 36 (87.8) | 60 (85.7) | 0.552 |
| • Normal | 5 (17.2) | 5 (12.2) | 10 (14.3) | |
| **Aganglionosis type** | | | | |
| • Short-segment | 27 (93.1) | 37 (90.2) | 64 (91.4) | 0.674 |
| • Long-segment | 2 (6.9) | 4 (9.8) | 6 (8.6) | |
| **Postoperative hemoglobin level** | | | | |
| • Anemia | 5 (17.2) | 12 (29.3) | 17 (24.3) | 0.347 |
| • Normal | 24 (86.8) | 29 (70.7) | 53 (75.7) | |
| **Postoperative albumin level** | | | | |
| • Hypoalbuminemia | 10 (34.5) | 16 (39) | 26 (37.1) | 0.698 |
| • Normal | 19 (65.5) | 25 (61) | 44 (62.9) | |
| **HAEC score** | | | | |
| • Cut off ≥ 4 | 2 (6.9) | 10 (24.4) | 12 (17.1) | 0.056 |
| • Cut off ≥ 10 | 1 (3.4) | 4 (9.8) | 5 (7.1) | 0.198 |
| **History** | | | | |
| • Diarrhea with an explosive stool | 2 (6.9) | 8 (19.5) | 10 (14.3) | 0.178 |
| • Diarrhea with foul-smelling stool | 2 (6.9) | 7 (17.1) | 9 (12.9) | 0.289 |
| • Diarrhea with bloody stool | 3 (10.3) | 6 (14.5) | 9 (12.9) | 0.726 |
| • History of enterocolitis | 3 (10.3) | 12 (41.3) | 15 (21.4) | 0.078 |
| **Physical examination** | | | | |
| • Explosive discharge of gas and stool on rectal examination | 1 (3.4) | 4 (9.8) | 5 (7.1) | 0.395 |
| • Abdominal distension | 6 (20.7) | 10 (24.3) | 16 (22.9) | 0.780 |
| • Decreased peripheral perfusion | 0 (0) | 3 (7.3) | 3 (4.3) | 0.261 |
| • Lethargy | 0 | 5 (12.2) | 5 (7.1) | 0.062 |
| • Fever | 2 (6.9) | 8 (19.5) | 10 (14.3) | 0.178 |
| **Radiologic examination** | | | | |
| • Multiple air-fluid levels | 1 (3.4) | 4 (9.8%) | 5 (7.1) | 0.395 |
| • Dilated bowel loops | 1 (3.4) | 6 (14.5%) | 7 (10) | 0.226 |
| • Sawtooth appearance with irregular mucosal lining | 0 | 2 (4.9%) | 2 (2.9) | 0.508 |
| • Cut-off sign in rectosigmoid with absence of distal air | 0 | 4 (9.8%) | 4 (5.7) | 0.136 |
| • Pneumatosis | 1 (3.4) | 4 (9.8%) | 5 (7.1) | 0.389 |
| **Laboratory** | | | | |
| • Leukocytosis | 2 (6.9) | 5 (12.2%) | 7 (10) | 0.691 |
| • Shift to left | 18 (62.1) | 19 (46.3%) | 37 (52.9) | 0.230 |

HAEC: Hirschsprung-associated enterocolitis; TSLPT: transanal Swenson-like pull-through; TEPT: transanal endorectal pull-through; [a]Analyzed using Chi-Square or Fisher Exact test.

## Association between predictor factors and Hirschsprung-associated enterocolitis in TSLPT and TEPT procedures

Next, we analyzed the association between predictor factors and HAEC in TSLPT and TEPT procedures using two scoring cut-off values (≥4 and ≥10) (Tables 3 and 4, respectively). Postoperative albumin level was the only variable that showed a significant association with HAEC in the TEPT group (p = 0.03). In contrast, other factors, including sex, age at surgery, nutritional status, and postoperative hemoglobin level, did not demonstrate significant associations with HAEC in either surgical group (Tables 3 and 4).

Significant post-hoc deviations were observed only for albumin in the TEPT ≥4 group, with hypoalbuminemia showing fewer-than-expected HAEC cases and normal albumin showing more. All other variables showed adjusted residuals <2 and were not significant (S2 Table).

## Multivariate analysis

On multivariate analysis, low albumin level remained significantly associated with HAEC occurrence after TEPT (p = 0.047; 95% CI: 0.010–0.970), although the wide confidence interval suggests a weak predictive value. Lower postoperative albumin levels were associated with increased odds of HAEC (OR = 0.096, 95% CI: 0.010–0.970, p = 0.047), indicating that decrease in albumin may increase the likelihood of HAEC. Other variables, including sex (OR = 0.245, 95% CI: 0.032–1.896), nutritional status (OR = 0.894, 95% CI: 0.056–14.353), and postoperative hemoglobin level (OR = 0.137, 95% CI: 0.009–2.038), did not show statistically significant associations with HAEC, and their wide confidence intervals suggest limited precision and a lack of strong predictive value (Table 5).

## Discussion

The proportion of HAEC following TSLPT and TEPT was analyzed using two scoring thresholds: ≥ 10 and ≥4. Using the ≥ 10 cut-off, the incidence was 3.4% in the TSLPT group and 9.8% in the TEPT group; when applying the ≥ 4 cut-off, the rates increased to 6.9% and 24.4%, respectively. TSLPT consistently showed lower incidence rates across both

**Table 3. Association between predictor factors and the occurrence of Hirschsprung-associated enterocolitis in the transanal Swenson-like pull-through group, analyzed using cut-off values of ≥4 and ≥10 (n = 29).**

| Characteristic | Cut-off value ≥4 | | | | Cut-off value ≥10 | | | |
|---|---|---|---|---|---|---|---|---|
| | HAEC | Non-HAEC | p-value[a] | OR (95% CI) | HAEC | Non-HAEC | p-value[a] | OR (95% CI) |
| **Sex** | | | | | | | | |
| Male | 1 | 18 | 0.632 | 1 | 1 | 18 | 0.460 | 1 |
| Female | 1 | 9 | | 2.00 (0.11–35.80) | 0 | 10 | | 0.58 (0.02–15.74) |
| **Age at surgery** | | | | | | | | |
| Neonates | 0 | 2 | 0.690 | 1 | 0 | 2 | 0.782 | 1 |
| Post-neonates | 1 | 25 | | 0.49 (0.01–13.32) | 1 | 26 | | 0.28 (0.00–0.894) |
| **Nutritional status** | | | | | | | | |
| Underweight | 2 | 22 | 0.504 | 1.22 (0.05–29.28) | 1 | 23 | 0.642 | 0.70 (0.02–19.66) |
| Normal | 0 | 5 | | 1 | 0 | 5 | | 1 |
| **Postoperative hemoglobin level** | | | | | | | | |
| Anemia | 0 | 5 | 0.504 | 0.81 (0.03–19.60) | 0 | 5 | 0.642 | 1.42 (0.05–39.89) |
| Normal | 2 | 22 | | 1 | 1 | 23 | | 1 |
| **Postoperative albumin level** | | | | | | | | |
| Hypoalbuminemia | 0 | 10 | 0.288 | 0.33 (0.01–7.63) | 1 | 13 | 0.698 | 0.58 (0.02–15.74) |
| Normal | 2 | 17 | | 1 | 0 | 15 | | 1 |

OR: odds ratio; CI; confidence interval; [a]Analyzed using the Chi-Square test; *Statistically significant at p < 0.05.

**Table 4. Association between predictor factors and the occurrence of Hirschsprung-associated enterocolitis in the transanal endorectal pull-through group, analyzed using cut-off values of ≥4 and ≥10 (n = 41).**

| Characteristic | Cut-off value ≥4 | | | | Cut-off value ≥10 | | | |
|---|---|---|---|---|---|---|---|---|
| | HAEC | Non-HAEC | p-value[a] | OR (95% CI) | HAEC | Non-HAEC | p-value[a] | OR (95% CI) |
| **Sex** | | | | | | | | |
| Male | 8 | 21 | 0.459 | 1 | 4 | 25 | 0.627 | 1 |
| Female | 2 | 10 | | 0.11 (0.02–0.66) | 1 | 11 | | 0.56 (0.05–5.68) |
| **Age at surgery** | | | | | | | | |
| Neonates | 0 | 0 | – | 1 | 0 | 0 | – | 1 |
| Post-neonates | 10 | 31 | | 0.33 (0.00–17.87) | 5 | 36 | | 0.15 (0.00–8.40) |
| **Nutritional status** | | | | | | | | |
| Underweight | 9 | 27 | 0.807 | 1.33 (0.13–13.53) | 5 | 31 | 0.374 | 1.92 (0.09–39.89) |
| Normal | 1 | 4 | | 1 | 0 | 5 | | 1 |
| **Postoperative hemoglobin level** | | | | | | | | |
| Anemia | 1 | 11 | 0.124 | 0.20 (0.02–1.81) | 1 | 10 | 0.713 | 0.65 (0.06–6.54) |
| Normal | 9 | 20 | | 1 | 4 | 26 | | 1 |
| **Postoperative albumin level** | | | | | | | | |
| Hypoalbuminemia | 1 | 15 | 0.030* | 0.11 (0.01–1.05) | 1 | 15 | 0.352 | 0.35 (0.03–3.45) |
| Normal | 9 | 16 | | 1 | 4 | 21 | | 1 |

OR: odds ratio; CI; confidence interval; [a]Analyzed using the Chi-Square test; *Statistically significant at p < 0.05.

**Table 5. Multivariate analysis of predictor factors associated with the occurrence of HAEC using a cut-off score ≥4 in the transanal endorectal pull-through (TEPT) group.**

| Variable | Odds Ratio (OR) | 95% Confidence Interval | p-value |
|---|---|---|---|
| Sex | 0.245 | 0.032–1.896 | 0.178 |
| Nutritional status | 0.894 | 0.056–14.353 | 0.937 |
| Hemoglobin level | 0.137 | 0.009–2.038 | 0.149 |
| Albumin level | 0.096 | 0.01–0.97 | *0.047** |

*, significant with p-value of <0.05.

thresholds; the difference between the two surgical techniques nearly reached statistical significance, especially with the cut-off of ≥4. These findings suggest a possible clinical advantage of TSLPT in minimizing the occurrence of HAEC, consistent with a previous report that has documented favorable outcomes and lower rates of HAEC following TSLPT [14]. Overall, the incidence of HAEC within the first six months postoperatively was 14% [14]. Variations in the reported rates of HAEC across different studies might be explained by the heterogeneity of diagnostic criteria and scoring systems utilized to define HAEC [9]. Differences in threshold scores, clinical parameters assessed, and the subjective interpretation of symptoms such as abdominal distension, fever, diarrhea, and leukocytosis contribute to inconsistencies in diagnosis. We applied two cut-off values (≥10 and ≥4) for HAEC diagnosis in our cohort. Prior studies have demonstrated that the higher threshold (≥10) yields high specificity but may miss milder cases [18], whereas the lower threshold (≥4) improves sensitivity and may capture earlier or milder episodes of HAEC [19,20]. By reporting both thresholds, we highlight how the incidence/proportion of HAEC varies depending on the diagnostic criteria used, and we recommend that future studies clearly state the cut-off used and validate scoring systems in their own populations.

In addition to procedural comparisons, several predictor factors were examined. Sex did not influence the proportion of HAEC. This aligns with several reports indicating that although HSCR is more common in males, sex itself does not seem to affect the risk of postoperative HAEC. A recent national cohort study exploring potential sex differences in HSCR outcomes found that the early clinical course of patients does not appear to depend on sex, with no significant difference in HAEC incidence between males and females [26]. Other literature reviews have similarly concluded that while some earlier, smaller studies suggested a possible link to female gender, the majority of evidence, including recent comprehensive analyses, points toward no significant association between sex and the risk of developing HAEC [27].

Similarly, neither the extent of aganglionosis nor the age at the time of definitive surgery showed a significant impact on the incidence of HAEC. A previous study has proposed that delayed surgical intervention might contribute to a more severe course of HAEC [7]. Prolonged retention of the aganglionic segment is thought to exacerbate intestinal dysmotility, leading to stasis of luminal contents and creating an environment favorable for bacterial overgrowth [7]. The resulting bacterial proliferation increases the risk of bacterial translocation across the compromised intestinal mucosa [7]. In addition, persistent dysmotility and accumulation of intraluminal contents can cause progressive bowel distension, which in turn elevates intramural pressure and impairs mucosal blood flow [7]. This sequence of events may culminate in intestinal ischemia, further compromising the integrity of the bowel wall and amplifying the inflammatory response characteristic of HAEC [7].

In this study, most patients with HAEC were underweight. This observation aligns with previous studies that underweight status might increase the risk of HAEC following colorectal procedures [9]. Optimal gastrointestinal function plays a crucial role in maintaining adequate growth, efficient nutrient absorption, robust immune responses, and protection against microbial translocation and systemic inflammation [28]. Impairments in these physiological processes may predispose underweight individuals to increased susceptibility to intestinal complications such as HAEC. Therefore, the application of appropriate and adequate surgical techniques is essential to preserve intestinal motility, minimize postoperative dysmotility, and reduce the risk of subsequent development of HAEC [28].

Anemia and hypoalbuminemia were also evaluated concerning surgical outcomes. Anemia was not significantly associated with either the surgical technique or the diagnostic cut-off value. Similarly, hypoalbuminemia showed no significant association, except in the TEPT group when the ≥4 cutoff was used ($p = 0.03$). These findings differ from an earlier study, which did not identify an association with albumin levels but reported hemoglobin concentration as a contributing factor in the TEPT group [9]. Other studies have demonstrated a significant association between HAEC and anemia, particularly microcytic hypochromic anemia [29]. Impaired bowel function characteristic of HAEC may lead to chronic gastrointestinal blood loss and disturbances in iron absorption, thereby exacerbating intestinal injury and inflammatory processes [29]. Anemia might develop at any point during the perioperative period, either as a result of pre-existing nutritional deficiencies or intraoperative blood loss [29]. Reduced hemoglobin levels can impair oxygen and nutrient delivery to intestinal tissues, resulting in compromised mucosal function and dysregulated immune responses, particularly under conditions of hypoxia [30]. Albumin plays a critical role in maintaining intestinal homeostasis, partly through its modulation of hypoxia-inducible factor-1α, a key regulator activated during hypoxic stress [31]. Deficiencies in both albumin and hemoglobin may impair mucosal barrier integrity and promote persistent inflammation, contributing to disease progression [30–32]. These findings emphasize the importance of optimizing physiological parameters, including hemoglobin and albumin levels, during both the preoperative and postoperative periods to support intestinal health and reduce the risk of adverse outcomes.

Among all evaluated variables in post-hoc analyses, albumin level was the only factor showing a statistically significant deviation in HAEC distribution, observed in the TEPT group at cut-off ≥4. Patients with normal albumin levels exhibited more HAEC events than expected, while hypoalbuminemic patients showed fewer. Although counterintuitive, this pattern may reflect selection bias or small sample behavior rather than a true biological effect, as the association was not reproduced in other cut-off levels or in the TSLPT group. No consistent significant findings were observed for sex, age at surgery, nutritional status, or hemoglobin level, suggesting that these variables may not contribute meaningfully to HAEC risk within the postoperative period.

The multivariate analysis demonstrated that postoperative hypoalbuminemia was the strongest risk indicator for HAEC in the TEPT group, with an odds ratio of 0.096 (95% CI: 0.010–0.970). This suggests that lower albumin may increase vulnerability to HAEC, although the wide confidence interval indicates moderate uncertainty in the magnitude of this effect. In contrast, sex, nutritional status, and postoperative hemoglobin levels showed no significant associations with HAEC, and their broad confidence intervals highlight limited predictive value. These findings underscore that while hypoalbuminemia may be clinically relevant in the TEPT cohort, the evidence does not support these other factors as meaningful predictors in our population.

The primary aim of this study was to analyze the difference in the incidence of HAEC following TEPT compared to TSLPT. The TEPT group demonstrated a tendency to have a higher incidence of HAEC compared to the TSLPT group. In the TEPT procedure, it is advisable to avoid the creation of an excessively long seromuscular cuff, as this might predispose to complications such as obstruction, constipation, constriction, and HAEC [33]. Hypoalbuminemia emerged as a significant predictor factor for HAEC in patients undergoing TEPT. Maintaining adequate albumin concentrations might contribute to improved intestinal integrity and immune function, thereby potentially reducing the incidence and severity of HAEC. Early identification and correction of hypoalbuminemia could serve as a practical strategy in postoperative care to minimize HAEC-related complications in these patients.

Several limitations must be acknowledged. The retrospective nature of this study limited the availability and completeness of medical records, particularly older data sets that often-lacked essential clinical details. The absence of comprehensive radiological findings might have affected the diagnostic accuracy of HAEC, potentially affecting the study's conclusions. The scoring system has limited prospective validation, and inter-rater reliability was not formally measured in our study. In addition, the relatively small sample size and involvement of multiple surgeons might have contributed to variability in surgical technique and outcomes, particularly in the TEPT group. Regular postoperative follow-up is recommended for patients with HSCR to monitor long-term outcomes and detect complications. Future research should focus on prospective data collection to enable more robust evaluation and the development of standardized algorithms for managing postoperative complications following TSLPT and TEPT.

## Conclusions

Our study shows that the incidence of HAEC is higher following TEPT than TSLPT, although this difference did not reach statistical significance. Notably, hypoalbuminemia appeared as a potential risk factor for HAEC in patients undergoing TEPT; however, given the limited statistical significance and confidence interval range, this association should be interpreted with caution. These findings highlight the importance of routine postoperative surveillance of nutritional status, particularly serum albumin levels, in children with HSCR.

## Supporting information

**S1 Table. Baseline characteristics and scoring system for Hirschsprung-associated enterocolitis patients treated with transanal soave longitudinal pull-through (TSLPT) and transanal endorectal pull-through (TEPT).**
(XLSX)

**S2 Table. Post-hoc Chi-square Adjusted Residuals for HAEC (H vs nH) Across TSLPT and TEPT Surgical Groups.**
(DOCX)

## Acknowledgments

We would like to extend our gratitude to the patients, parents, and everyone who participated in the study, providing excellent technical support and assistance. Some of the manuscript's results are from Azzahra Fatinnuha Azmi Prayogi Putri's thesis.

# Author contributions

**Conceptualization:** Kristy Iskandar, Eko Purnomo, Gunadi.

**Data curation:** Azzahra Fatinnuha Azmi Prayogi Putri, Dwiki Afandy, Pramana Adhityo.

**Formal analysis:** Azzahra Fatinnuha Azmi Prayogi Putri, Dwiki Afandy, Ahmad Zakiy Habibiy, Pramana Adhityo, Gilang Vigorous Akbar Eka Candy, Gunadi.

**Investigation:** Dwiki Afandy, Ahmad Zakiy Habibiy, Khanza Adzkia Vujira, Pramana Adhityo, Gunadi.

**Methodology:** Azzahra Fatinnuha Azmi Prayogi Putri, Setiani Silvy Nurhidayah.

**Project administration:** Setiani Silvy Nurhidayah, Khanza Adzkia Vujira, Gunadi.

**Resources:** Kristy Iskandar, Eko Purnomo, Gunadi.

**Supervision:** Kristy Iskandar, Eko Purnomo, Gunadi.

**Validation:** Gunadi.

**Visualization:** Pramana Adhityo, Gilang Vigorous Akbar Eka Candy.

**Writing – original draft:** Azzahra Fatinnuha Azmi Prayogi Putri, Dwiki Afandy, Pramana Adhityo, Gunadi.

**Writing – review & editing:** Azzahra Fatinnuha Azmi Prayogi Putri, Dwiki Afandy, Ahmad Zakiy Habibiy, Setiani Silvy Nurhidayah, Khanza Adzkia Vujira, Pramana Adhityo, Gilang Vigorous Akbar Eka Candy, Kristy Iskandar, Eko Purnomo, Gunadi.

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
