## [Decision Letter · Decision Letter 0]

7 Oct 2025

PONE-D-25-45077Does surgical approach affect Hirschsprung-associated enterocolitis risk? A comparison between transanal Swenson-like and endorectal pull-throughsPLOS ONE?

Dear Dr. Gunadi,

Thank you for submitting your manuscript to PLOS ONE. After careful consideration, we feel that it has merit but does not fully meet PLOS ONE’s publication criteria as it currently stands. Therefore, we invite you to submit a revised version of the manuscript that addresses the points raised during the review process.

We look forward to receiving your revised manuscript.

Kind regards,

Kota V Ramana, Ph.D.

Academic Editor

PLOS ONE

Journal Requirements:

Reviewers' comments:

Reviewer's Responses to Questions

**Comments to the Author**

1. Is the manuscript technically sound, and do the data support the conclusions?

Reviewer #1: Partly

Reviewer #2: No

2. Has the statistical analysis been performed appropriately and rigorously?

Reviewer #1: Yes

Reviewer #2: No

3. Have the authors made all data underlying the findings in their manuscript fully available?

Reviewer #1: Yes

Reviewer #2: No

4. Is the manuscript presented in an intelligible fashion and written in standard English?

Reviewer #1: Yes

Reviewer #2: Yes

Reviewer #1: Paper review

Does surgical approach affect Hirschsprung-associated enterocolitis risk? A comparison between transanal Swenson-like and endorectal pull-throughs

General statement:

The study by Azzahra et al, aimed to analyze the difference in the incidence of Hirschprug-associated enterocolitis following Transanal endorectal pull through (TEPT) compared to transanal Swenson-like pull-through (TSLPT). The application of Chi square test of independence/ Fisher’s exact test, and Multivariate analysis (logistic regression) to identify independent predictors of HAEC are appropriate for this study.

However, the following comments should be taken note of

Comments/ Clarifications:

Abstract: This section needs to be re-written

1) The statement, “Hirschsprung-associated enterocolitis (HAEC) is a complication of Hirschsprung disease (HSCR) that occurs before and after surgery.” May be changed to “Hirschsprung-associated enterocolitis (HAEC) is a complication of Hirschsprung disease (HSCR) that may occur both before or after surgery”

2) The statement, “We aimed to compare the incidence of HAEC following TSLPT and TEPT in HSCR patients...” may better read: “We aimed to compare the incidence of HAEC following TSLPT versus TEPT in HSCR patients…”

3) The statement, “We retrospectively reviewed the medical records of HSCR patients who underwent TSLPT and TEPT at our institution between 2018 and 2023” may better read: “We retrospectively reviewed the medical records of HSCR patients who underwent either TSLPT or TEPT at our institution between 2018 and 2023”

4) The word “frequency” in this statement and in several other places in the manuscript, have been used interchangeably with “incidence” which promotes ambiguity: The frequency of HAEC in TSLPT and TEPT was 24.4% and 6.9%, respectively. The authors need to clarify this. Most times “Frequency” is used to denote “absolute frequency” which is presented as counts. ‘Relative frequency’ is a fraction of the total which the authors maybe trying to present. ‘Proportion’ is still an appropriate term.

5) The authors need to take note of this statement written in the abstract, results etc: “These differences almost reached a significant level (p=0.056)”. It is necessary for the authors to state that the difference between the incidence of HAEC in TSLPT vs TEPT did not reach statistical significance. The statement, “almost reached a significant level (p=0.056)” may not be scientifically acceptable and its interpretation should be that of an insignificant finding.

6) The use of “prognostic factors” in the abstract and other sections of the manuscript is confusing. Prognostic factors are factors that predict the long term outcome/progression of a disease. Do the authors rather mean risk/predictor factors, which are more appropriate terms. If the authors actually mean prognostic factors then they should discuss more about these prognostic factors in the introduction/ discussion section. The authors should note the difference between risk factors, predictive factors, associated factors, prognostic factors and apply the terminology appropriately.

7) This statement written in the abstract and in line 193: “Subsequently, multivariate analysis indicated that albumin level is a strong prognostic factor for HAEC in HSCR patients following TEPT (p=0.047).” does not seem to be completely true, judging by the result obtained and presented in table 2. The confidence interval is showing it may rather be a weak predictor (the CI is wide and very close to 1.0)

8) These two sentences in the abstract are contradictory: “The frequency of HAEC in TSLPT and TEPT was 24.4% and 6.9%, respectively.” Vs “In conclusion, our study suggests that the incidence of HAEC tends to be higher following TEPT compared to TSLPT.”

9) This statement: “Furthermore, it emphasizes the importance of routinely monitoring albumin levels postoperatively to prevent HAEC in children with HSCR”, appears to over-generalize since the albumin levels was only significant in the TEPT group.

Introduction:

10) Since HAEC is the central outcome of interest, a bit more discussion of its pathophysiology or why it differs between techniques would be appropriate.

Materials and Methods:

11) Lines 89-90: “Data on HAEC in patients who underwent TSLPT and TEPT between 2018 and 2023...” may better read: “Data on HAEC in patients who underwent TSLPT or TEPT between 2018 and 2023…”

12) Two cut-off values, i.e., ≥4 and ≥10, were applied. The authors should indicate clearly what these cut-offs mean.

13) The authors need to clarify this sentence: Before participating, each patient's parent completed a written informed consent form. Was the consent obtained before the surgeries or obtained for the sole purpose of the present study.

14) “The data were accessed for research purposes from 02/10/2023 to 02/02/2024” This should better be taken to the data collection section and clarified as such.

15) The sentence: “Authors had access to information that could identify individual participants” may raise ethical concerns unless you clarify how privacy was protected (e.g., anonymized before analysis).

16) Lines 130-132: Nutritional status was assessed based on body mass index (BMI) and categorized as underweight (95th percentile) should be brought up close to other demographics ie after age classification on line 126.

17) Also, the BMI chart that was used for the percentile classification should be indicated. For example, CDC chart, WHO chart.

18) Table 1 and 2: “Diarrhea with an explosive tool.” Do the authors mean “Diarrhea with an explosive stool” instead.

Sample Size and Sampling Method:

19) The sample size calculation does not seem to be appropriate. The calculation using the equation presented, gives 47.4 instead of 68.

20) The authors wrote, “Substituting these values into the formula yielded… resulting in an estimated sample size of 68.” The authors however used sample size of 70. An explanation should be added. For example, the calculated sample size was 47 however all eligible cases were included which yielded a final sample size of 70. Whatever value is presented in this section should tally with the final sample size used.

21) “Chi-square or Fisher’s exact tests was applied to compare the frequency of HAEC between the TSLPT and TEPT groups”. This is vague. The authors should state the conditions where either Chi-square or Fisher’s exact tests were applied.

Results:

22) Table 2,3 For the last column p-valuea, the authors may need to indicate (a) beside ones that were Fisher’s exact or if they choose, beside ones for chi square, and label beneath the table what the ones with (a) indicate, not both tests.

23) Authors should consider reorganizing table 2 for better flow. Demographics like sex, age and nutritional status should come before Aganglionosis type and other factors.

24) The authors wrote: “The frequency of HAEC in TSLPT and TEPT was 24.4% and 6.9%, respectively, using a cut-off of ≥4.” What about the proportion of HAEC in TSLPT and TEPT using a cut-off of ≥10 for completion?

25) In reporting the findings on line 179-182, authors should present the significant finding before the insignificant ones.

26) Looking at table 3 for instance, the total numbers for each of the variables do not round up to 70 as compared to table 1. For example, the variable “sex” total number in the Cut-off value ≥4 and Cut-off value ≥10 for both HAEC and Non-HAEC in the males and females categories is = 1 +18+ 1+ 9 + 1 + 18 + 0+ 10 = 58 and not 70. This is same for other variables. Authors should clarify the inconsistency in the total values.

27) Authors should interpret the multivariate analysis (logistic regression) using the odds ratio and confidence interval (i.e. risk). There was no interpretation of this in the entire manuscript. It may not be so appropriate to interpret logistic regression using p value alone.

Discussion:

28) Line 257 - 258: “The primary aim of this study was to evaluate the most appropriate surgical technique for the early prevention and management of HAEC”. This statement may not be appropriate as it is different from the primary aim of the present study which is “to analyze the difference in the incidence of Hirschprug-associated enterocolitis following Transanal endorectal pull through (TEPT) compared to transanal Swenson-like pull-through (TSLPT).”

29) Line 280 under Conclusions: Our study shows that the trend of HAEC is higher following TEPT than TSLPT. The authors should note that the present study did not examine trend, but rather examined the incidences.

30) Additionally, if they state the incidence of HAEC is higher following TEPT than TSLPT, they should add, “although it did not reach statistical significance.

31) Because hypoalbuminemia was only significant for the TEPT group at the cut-off of ≥4, it should be recognized as a potential risk factor not a definite prognostic factor.

32) Under the References section; Authors should cross-check the referencing style for appropriate use of upper case(Capital letters) in the titles and Journal names.

33) Additionally, the references should maintain same style. For example, to match this style: 2014;9(5):264-269 (if it is the chosen/accepted style)

Reference 5: 2022;17(1):150–4.

Reference 7: 2003; 238(4):569–76

Reference 22: 2023;15;10:1055128.

Reviewer #2: Dear author

I have reviewed the manuscript titled “Does surgical approach affect Hirschsprung-associated enterocolitis risk? A comparison between transanal Swenson-like and endorectal pull-throughs" (PONE-D-25-45077).

There is some important comment that should be considered.

1. The introduction cites recent studies from 2023 and 2024 (references 9 and 10) suggesting that the transanal Swenson-like pull-through (TSLPT) technique may reduce mechanical and functional complications, including Hirschsprung-associated enterocolitis (HAEC). However, immediately following this, the manuscript references a much older study from 2011 (reference 11) to emphasize that despite surgical advances, HAEC remains a significant risk for morbidity and mortality. This sequence creates a temporal disconnect in the justification of the study, as the more recent references imply ongoing advancements, while the older reference suggests persistent high risk.

2. Additionally, the manuscript cites an older reference from 1989 (reference 12) to support the claim that neonates with Hirschsprung disease who develop HAEC experience hospital stays twice as long as those without this complication. Given the age of this reference, it raises concerns about the currency and contextual relevance of this justification, especially when more recent studies have likely addressed hospital stay durations and morbidity associated with HAEC. The authors should consider including updated evidence to maintain the temporal consistency and strengthen the rationale of their study

3. Based on the relevance of the prognostic factors analyzed—sex, aganglionosis type, nutritional status, age at surgery, and postoperative hemoglobin and albumin levels—it would strengthen the manuscript if the authors explicitly addressed and discussed these factors in the discussion section. Highlighting their importance and contextualizing the study findings within the framework of existing literature on these predictive factors would provide a clearer interpretation of the results and reinforce the study’s significance

4. The authors have applied appropriate statistical methods, including sample size calculation based on a single-proportion formula and suitable use of Chi-square, Fisher’s exact, and logistic regression analyses. However, the relatively small sample sizes within each surgical subgroup (29 in TSLPT, 41 in TEPT) may limit the power to detect subtle differences, raising the possibility of type II error. It would strengthen the manuscript if the authors explicitly acknowledge this limitation and consider providing post-hoc power analyses to better contextualize non-significant results that may still be clinically relevant. Furthermore, a more detailed description of how missing data were handled and whether any sensitivity analyses were performed would improve confidence in the robustness of the findings. Overall, while the statistical approach is sound, caution should be exercised in interpreting results, particularly borderline findings, given the sample size and retrospective design constraints.

5. The manuscript's description of the HAEC scoring system lacks sufficient detail about the specific diagnostic criteria, including how the scores are determined and whether the scoring has been validated or has good reproducibility. As the diagnosis of HAEC is inherently challenging due to its non-specific signs and symptoms, providing a clear and detailed explanation of the scoring system would enhance the transparency and reproducibility of the study. Currently, the manuscript would benefit from elaborating on how the scores were assigned, who performed the scoring, and whether any measures were taken to ensure consistency across different raters or centers.

**Do you want your identity to be public for this peer review?** For information about this choice, including consent withdrawal, please see our Privacy Policy

Reviewer #1: No

Reviewer #2: No

---

## [Author Response · Author response to Decision Letter 1]

4 Dec 2025

Response to Reviewers and Editors

Manuscript (Revised) Title: Does surgical approach affect Hirschsprung-associated enterocolitis risk? A comparison between transanal Swenson-like and endorectal pull-throughs

Reference #: Submission ID PONE-D-25-45077

Date:

Dear Editorial Office, PLOS One,

We are grateful for the chance to revise our manuscript based on the valuable feedback from the reviewers and editors. We appreciate your constructive comments, which have enabled us further to enhance the quality and clarity of our paper. We have thoroughly addressed each point and revised the manuscript accordingly. The changes are listed in the revision section below.

Please find our point-by-point responses to the comments, with files consisting of:

● A marked-up copy of the manuscript that highlights changes made to the original version labeled 'Revised Manuscript with Track Changes'.

● An unmarked version of the revised paper without tracked changes, labeled 'Manuscript'.

Editorial Comments: When submitting your revision, we need you to address these additional requirements.

1. Comment: Please ensure that your manuscript meets PLOS ONE's style requirements, including those for file naming. The PLOS ONE style templates can be found at

○ Response: Thank you for this important reminder. We have carefully reviewed and revised the manuscript to ensure full compliance with PLOS ONE’s formatting and submission requirements. This includes adjustments to title page structure, author affiliations, heading levels, reference formatting, and figure/table legends according to the official PLOS ONE style templates. We have also ensured that all submitted files follow the required PLOS ONE file naming conventions.

○ Changes: The manuscript and all associated submission files (main text, figures, tables, and supplementary materials) have been reformatted to conform with PLOS ONE’s style and file-naming guidelines as outlined in the provided templates.

2. Comment: Please provide additional details regarding participant consent. In the ethics statement in the Methods and online submission information, please ensure that you have specified (1) whether consent was informed and (2) what type you obtained (for instance, written or verbal, and if verbal, how it was documented and witnessed). If your study included minors, state whether you obtained consent from parents or guardians. If the need for consent was waived by the ethics committee, please include this information.

○ Response: Thank you for this important comment. We have revised the Ethics Statement in the Methods section and the online submission form to provide detailed information regarding participant consent. Written informed consent was obtained from the parents or legal guardians of all pediatric participants for the use of their anonymized medical data for research purposes. The consent process was conducted in accordance with institutional and ethical standards, and all data were fully anonymized prior to analysis.

○ Changes: The protocol of this retrospective study was reviewed and approved by the Medical and Health Research Ethics Committee (MHREC), Faculty of Medicine, Public Health, and Nursing, Universitas Gadjah Mada, Yogyakarta, Indonesia (Approval number: KE/FK/0937/EC). Written informed consent was obtained from the parents or legal guardians of all participating patients, permitting the use of anonymized medical data for research purposes. All data were fully anonymized before access and analysis. The study was conducted in accordance with the principles of the Declaration of Helsinki and institutional ethical guidelines.

3. Comment: Your ethics statement should only appear in the Methods section of your manuscript. If your ethics statement is written in any section besides the Methods, please delete it from any other section.

○ Response: Thank you for the clarification. We have revised the manuscript accordingly. The Ethics Statement and Consent Information have been moved from the section after the Conclusion to the Methods section, in compliance with PLOS ONE’s formatting and reporting requirements. No ethical or consent information now appears outside of the Methods section.

○ Changes: The protocol of this retrospective study was reviewed and approved by the Medical and Health Research Ethics Committee (MHREC), Faculty of Medicine, Public Health, and Nursing, Universitas Gadjah Mada, Yogyakarta, Indonesia (Approval number: KE/FK/0937/EC). Written informed consent was obtained from the parents or legal guardians of all participating patients, permitting the use of anonymized medical data for research purposes. All data were fully anonymized before access and analysis. The study was conducted in accordance with the principles of the Declaration of Helsinki and institutional ethical guidelines.

4. Comment: If the reviewer comments include a recommendation to cite specific previously published works, please review and evaluate these publications to determine whether they are relevant and should be cited. There is no requirement to cite these works unless the editor has indicated otherwise.

○ Response: Thank you for the reminder. We have carefully reviewed all the suggested references provided by the reviewers. Relevant and appropriate citations have been incorporated into the manuscript to strengthen the background and discussion sections. At the same time, non-relevant works have been excluded to maintain focus on our study objectives. All newly added citations have been formatted in accordance with PLOS ONE reference style requirements.

Reviewer #1 Comments:

Does surgical approach affect Hirschsprung-associated enterocolitis risk? A comparison between transanal Swenson-like and endorectal pull-throughs

General statement:

The study by Azzahra et al. aimed to analyze the difference in the incidence of Hirschsprung-associated enterocolitis following Transanal endorectal pull-through (TEPT) compared to transanal Swenson-like pull-through (TSLPT). The application of Chi square test of independence/ Fisher’s exact test, and Multivariate analysis (logistic regression) to identify independent predictors of HAEC are appropriate for this study.

However, the following comments should be taken note of:

Abstract: This section needs to be rewritten:

1. The statement, “Hirschsprung-associated enterocolitis (HAEC) is a complication of Hirschsprung disease (HSCR) that occurs before and after surgery.” May be changed to “Hirschsprung-associated enterocolitis (HAEC) is a complication of Hirschsprung disease (HSCR) that may occur both before or after surgery”

ü Response: Thank you for the valuable comment. We have revised the sentence to improve clarity and to reflect better that Hirschsprung-associated enterocolitis (HAEC) can occur either before or after surgical treatment.

ü Changes: Hirschsprung-associated enterocolitis (HAEC) is a complication of Hirschsprung disease (HSCR) that may occur both before or after surgery.

2. The statement, “We aimed to compare the incidence of HAEC following TSLPT and TEPT in HSCR patients...” may better read: “We aimed to compare the incidence of HAEC following TSLPT versus TEPT in HSCR patients…”

ü Response: Thank you for your helpful suggestion. We agree with this recommendation and have revised the sentence accordingly to enhance clarity and readability.

ü Changes: We aimed to compare the incidence of HAEC following TSLPT versus TEPT in HSCR patients.

3. The statement, “We retrospectively reviewed the medical records of HSCR patients who underwent TSLPT and TEPT at our institution between 2018 and 2023” may better read: “We retrospectively reviewed the medical records of HSCR patients who underwent either TSLPT or TEPT at our institution between 2018 and 2023”

ü Response: Thank you for this valuable suggestion. We agree with the reviewer’s comment and have revised the sentence to accurately indicate that each patient underwent one of the two surgical procedures.

ü Changes: We retrospectively reviewed the medical records of HSCR patients who underwent either TSLPT or TEPT at our institution between 2018 and 2023.

4. The word “frequency” in this statement and in several other places in the manuscript, have been used interchangeably with “incidence” which promotes ambiguity: The frequency of HAEC in TSLPT and TEPT was 24.4% and 6.9%, respectively. The authors need to clarify this. Most times “Frequency” is used to denote “absolute frequency” which is presented as counts. ‘Relative frequency’ is a fraction of the total which the authors may be trying to present. ‘Proportion’ is still an appropriate term.

ü Response: Thank you very much for this valuable suggestion. We agree with the reviewer’s comment and have revised the term “frequency” to “proportion” throughout the manuscript to more accurately describe the proportion of HAEC cases among patients in each surgical group. This change improves clarity and aligns with appropriate epidemiological terminology.

5. The authors need to take note of this statement written in the abstract, results etc: “These differences almost reached a significant level (p=0.056)”. It is necessary for the authors to state that the difference between the incidence of HAEC in TSLPT vs TEPT did not reach statistical significance. The statement, “almost reached a significant level (p=0.056)” may not be scientifically acceptable and its interpretation should be that of an insignificant finding.

ü Response: Thank you very much for this important clarification. We agree with the reviewer’s comment. The wording has been revised throughout the manuscript (including in the Abstract and Results sections) to reflect that the difference was not statistically significant accurately.

ü Changes:

Abstract: “There was no statistically significant difference in the incidence of HAEC between the TSLPT and TEPT groups (p=0.056).”

Results: “Although the incidence of HAEC was higher in the TSLPT group compared to the TEPT group, the difference did not reach statistical significance (p=0.056).”

6. The use of “prognostic factors” in the abstract and other sections of the manuscript is confusing. Prognostic factors are factors that predict the long-term outcome/progression of a disease. Do the authors rather mean risk/predictor factors, which are more appropriate terms? If the authors actually mean prognostic factors then they should discuss more about these prognostic factors in the introduction/ discussion section. The authors should note the difference between risk factors, predictive factors, associated factors, prognostic factors and apply the terminology appropriately.

ü Response: Thank you for this insightful and constructive comment. We agree with the reviewer’s observation. In our study, the intent was to identify predictor factors associated with the occurrence of HAEC, not to evaluate long-term outcomes. Therefore, we have replaced the term “prognostic factors” with “predictor factors” throughout the manuscript to reflect the study objective more accurately.

7. This statement is written in the abstract and in line 193: “Subsequently, multivariate analysis indicated that albumin level is a strong prognostic factor for HAEC in HSCR patients following TEPT (p=0.047).” does not seem to be completely true, judging by the result obtained and presented in table 2. The confidence interval is showing it may rather be a weak predictor (the CI is wide and very close to 1.0)

ü Response: Thank you very much for this important observation. We agree with the reviewer’s assessment. The wording has been revised to accurately reflect the statistical findings and to avoid overstating the strength of association. The revised version now indicates that albumin level was significantly associated with HAEC rather than being described as a strong prognostic factor. Additionally, we have replaced “prognostic factor” with “risk factor” throughout the manuscript, in accordance with an earlier reviewer comment.

ü Changes:

Abstract: “Multivariate analysis indicated that low albumin level was significantly associated with the occurrence of HAEC in HSCR patients following TEPT (p=0.047).”

Results: “On multivariate analysis, low albumin level remained significantly associated with HAEC occurrence after TEPT (p=0.047; 95% CI: 0.010–0.970), although the wide confidence interval suggests a weak predictive value.”

8. These two sentences in the abstract are contradictory: “The frequency of HAEC in TSLPT and TEPT was 24.4% and 6.9%, respectively.” Vs “In conclusion, our study suggests that the incidence of HAEC tends to be higher following TEPT compared to TSLPT.”

ü Response: Thank you for your valuable comment. We appreciate your careful reading of the abstract. The inconsistency you noted resulted from a typographical error in the reported proportions of HAEC between the TSLPT and TEPT groups. The correct statement should read: “The proportion of HAEC in TEPT and TSLPT was 24.4% and 6.9%, respectively.” We have corrected this error in the revised manuscript. With this correction, the conclusion remains accurate and is supported by the corrected data.

ü Changes: The proportion of HAEC in TEPT and TSLPT was 24.4% and 6.9%, respectively.

9. This statement: “Furthermore, it emphasizes the importance of routinely monitoring albumin levels postoperatively to prevent HAEC in children with HSCR”, appears to over-generalize since the albumin levels was only significant in the TEPT group.

ü Response: Thank you for this helpful comment. We agree that the original sentence over-generalized the role of albumin. Since hypoalbuminemia was found to be significant only within the TEPT subgroup, we have revised the statement to accurately reflect that postoperative albumin monitoring may serve as a potential risk indicator for HAEC specifically in TEPT patients, rather than a universal factor for all HSCR patients.

ü Changes: Monitoring albumin levels postoperatively may be considered as a potential risk indicator for diagnosing HAEC in patients undergoing surgery especially TEPT, as hypoalbuminemia showed significance only within this surgical subgroup.

Introduction:

10. Since HAEC is the central outcome of interest, a bit more discussion of its pathophysiology or why it differs between techniques would be appropriate.

ü Response: Thank you for this insightful suggestion. We agree that expanding the discussion on the pathophysiology of HAEC and its potential relationship with different surgical techniques will improve the clarity and context of our findings. We have added a more detailed explanation in the Discussion section addressing the underlying mechanisms—such as impaired enteric nervous system function, altered gut motility, dysbiosis, and mucosal immune dysregulation—and how different pull-through techniques may influence these factors. This addition strengthens the rationale for why HAEC incidence may vary between TSLPT and TEPT.

ü Changes: HAEC is hypothesized to arise from a multifactorial interplay of decreased enteric nervous system function, altered gut motility, dysbiosis, and dysregulation of mucosal immunity [8,16,17]. Variations in surgical techniques may influence these parameters differently. For example, differences in the extent of aganglionic segment excision, anastomotic tension, and postoperative colonic motility restoration may contribute to the varying susceptibility to HAEC between procedures [16,17]. These physiological distinctions may partly explain the differing proportions of HAEC observed between TSLPT and TEPT in our cohort.

ü References:

8.Gershon EM, Rodriguez L, Arbizu RA. Hirschsprung's disease

---

## [Decision Letter · Decision Letter 1]

29 Dec 2025

Does surgical approach affect Hirschsprung-associated enterocolitis risk? A comparison between transanal Swenson-like and endorectal pull-throughs

PONE-D-25-45077R1

Dear Dr. Gunadi,

We’re pleased to inform you that your manuscript has been judged scientifically suitable for publication and will be formally accepted for publication once it meets all outstanding technical requirements.

Kind regards,

Kota V Ramana, Ph.D.

Academic Editor

PLOS One

Additional Editor Comments (optional):

Reviewers' comments:

Reviewer's Responses to Questions

**Comments to the Author**

Reviewer #1: All comments have been addressed

Reviewer #2: All comments have been addressed

2. Is the manuscript technically sound, and do the data support the conclusions?

Reviewer #1: Yes

Reviewer #2: Partly

3. Has the statistical analysis been performed appropriately and rigorously?

Reviewer #1: Yes

Reviewer #2: Yes

4. Have the authors made all data underlying the findings in their manuscript fully available?

Reviewer #1: Yes

Reviewer #2: Yes

5. Is the manuscript presented in an intelligible fashion and written in standard English?

Reviewer #1: Yes

Reviewer #2: Yes

Reviewer #1: (No Response)

Reviewer #2: The authors have adequately addressed all previous review comments, implementing changes to terminology, statistical interpretation, ethics/consent details, HAEC scoring, sample size justification, table reorganization, and references. Key strengths include accurate Abstract/Results reporting of non-significant HAEC differences, chronological Discussion reorganization, and hypoalbuminemia framed.

**Do you want your identity to be public for this peer review?** For information about this choice, including consent withdrawal, please see our Privacy Policy

Reviewer #1: No

Reviewer #2: No

---

## [Editor Report · Acceptance letter]

PONE-D-25-45077R1

PLOS One

Dear Dr. Gunadi,

I'm pleased to inform you that your manuscript has been deemed suitable for publication in PLOS One. Congratulations! Your manuscript is now being handed over to our production team.

Kind regards,

on behalf of

Dr. Kota V Ramana

Academic Editor

PLOS One